# Western Hognose Snakes (*Heterodon nasicus*) Prefer Environmental Enrichment

**DOI:** 10.3390/ani12233347

**Published:** 2022-11-29

**Authors:** Gokulan Nagabaskaran, Morgan Skinner, Noam Miller

**Affiliations:** Department of Psychology, Wilfrid Laurier University, 75 University Ave West, Waterloo, ON N2L 3C5, Canada

**Keywords:** western hognose snake, environmental enrichment, preference, reptile welfare, boldness

## Abstract

**Simple Summary:**

With the growing popularity of snakes in the pet trade, investigation into environmental enrichment preferences in snakes is critical to their captive welfare. Unfortunately, small, minimalistic enclosures are predominantly used by both pet owners and breeders alike to house captive snakes. Recently, a handful of studies have shown the benefits that arise from the correct use of enrichment with climbing snakes, but none have observed this aspect in popular pet snakes that burrow, such as western hognose snakes. This study found that hognose snakes significantly prefer environmental enrichment when given a choice between enrichment and minimalistic conditions. This encourages further investigation into the welfare needs of other captive snake species that may be improperly housed, leading to chronic stress and poor health.

**Abstract:**

The environmental enrichment needs of snakes are often disregarded. Using preference testing, we aimed to shed light on the enrichment preferences of a popular pet species, the western hognose snake (*Heterodon nasicus*). Snakes’ enclosures were divided into enriched and standard sides. The enriched half had substrate for burrowing, interactive stimuli, and a large water dish. The standard half had paper towel substrate and a small water dish. Each side also contained a single shelter. We provided belly heat to create a thermal gradient on one side of the cage. Snakes were observed for 6 days, four times daily. We predicted a preference for enriched conditions and, as snakes are ectothermic, a preference for the warmer side. Snakes were additionally given an exploration assay, to explore whether differences in preference for environmental enrichment interact with boldness levels. We found that hognose snakes preferred enrichment, and the strength of this preference increased over time. Preference for enrichment was stronger when the enriched side was cooler. This may be due to the burrowing tendencies of these snakes. We found no relationship between preference and boldness. These findings emphasise the importance of preference testing in establishing research-informed enrichment opportunities for reptiles.

## 1. Introduction

Understanding an animal’s environmental requirements is critical to its well-being when under human care, and can also inform conservation efforts. Unfit housing stemming from a lack of basic needs and environmental enrichment (EE) has been shown to be detrimental to captive animals and can impair their health [1]. This is commonly attributed to undue stress, which can arise from multiple aspects of deficient housing, and that may be mitigated with the implementation of EE. For example, inadequate enclosure size restricts the animal’s ability to engage in natural behaviours and increases physiological stress markers [2]. Lack of basic necessities allowing for species-typical behaviours also increases stress, for example, when providing inadequate flooring for burrowing species [3]. Thus, understanding the complete behavioural repertoire and needs of the animal in question is necessary to maintain healthy individuals in captivity, and EE can aid in this.

EE is commonly defined as the inclusion of naturalistic stimuli that promote natural behaviours, such as biologically relevant objects or conspecifics [1]. The benefits of EE have been well-documented in mammals and birds [4,5]. Unfortunately, the EE preferences of less popular species, such as snakes and other reptiles, are rarely studied [6,7,8]. This is problematic as the number of snakes under human care are increasing, particularly as pets [9]. The importance of expanding the literature on reptile enrichment is further emphasised by those few studies that have shown its benefits. EE produces cognitive benefits in turtles [10], lizards [7,11], and snakes [9,12]. For example, rat snakes (*Pantherophis obsoleta*) were housed under either standard or enriched conditions [12]. Enriched snakes were provided with shelters varying in elevation and humidity, aspen substrate, and live feedings, while snakes in standard enclosures were given a single shelter and water dish, corrugated paper substrate and dead prey. Enriched snakes grew larger than their standard housed conspecifics and demonstrated both better performance on a problem-solving task and faster habituation to an open field [12].

In another study, some corn snakes (*Pantherophis guttatus*) were provided with EE, including aspen bedding, branches and a board on which to climb, a large water dish, and various hides [9]. Snakes housed under standard conditions were given whole newspaper bedding with a single shelter and a small water dish. In as little as one-month, enriched snakes displayed the ability to discriminate familiar handlers from strangers by chemoreception, an ability that was absent in standard housed snakes [9]. Ball pythons (*Python regius*) have also been exposed to both EE and standard conditions, each for an eight-week period [13]. The EE took the form of soil substrate, climbing enrichment via branches and plants, as well as an abundance of natural components such as bark, grass, moss, and roots. The standard enclosures only had whole newspaper bedding, a single shelter and a single small water dish. When in enrichment, snakes showed significantly reduced stereotypical behaviours and engaged in more natural behaviours compared to their time in standard enclosures [13]. Thus, when implemented properly, the cognitive benefits of EE in snakes are substantial and observable after a short time.

An important first step to implementing proper enrichment, for any species, is preference testing [14], since what counts as an enriched environment is species specific. Preference testing is an attempt to demonstrate that animals prefer one type of housing or resource over another. Preference testing has been used to ascertain the enrichment preferences of numerous species, including rats, hens, and mink [15,16]. The only preference study we are aware of involving snakes [8] demonstrated that semi-arboreal corn snakes significantly preferred enriched environments when they could freely access both enriched and standard enclosures. However, the enrichment provided to the corn snakes would likely not be ideal for other snake species, due to the wide ecological diversity of snakes [17].

Another component of EE, rarely studied in reptiles, that may impact its effectiveness is personality. Personality is defined as individual differences in behaviour that are consistent across contexts and/or time [18]. In non-human animals, two examples of commonly studied traits are boldness and sociability [19,20]. The potential role of personality traits in welfare and conservation is of increasing interest [21]. The few existing studies of personality in snake species (*Boa imperator* [22]; *Thamnophis sirtalis sirtalis* [23,24,25]; *Crotalus oreganus* [26]) suggest that snakes of several species display consistent differences and that these internal biases have important effects on their behaviour, and may contain important lessons for their management and conservation [26]. Understanding the relationship between personality and enrichment may provide valuable insight into individual differences in enrichment use and preference. For example, if shy individuals avoid novel EE, enrichment could be introduced slowly to allow shy individuals to acclimate.

In the current experiment, in addition to preference for physical EE, we also explored snakes’ preference for heat. As snakes are ectothermic, they require access to environmental sources for thermoregulation. To allow snakes to regulate their internal temperature, provision of a heat gradient in their housing is often recommended, especially since different species may prefer different temperatures, and preferences may change when snakes are shedding or digesting a meal [27]. We therefore provided all our snakes with a heat gradient emanating from one corner of their cages (see Methods for details). Since warmer areas may be attractive and the presence of heat might therefore be confounded with enrichment preferences, we counterbalanced whether the warmer side was the enriched or standard side of the cage. Counterbalancing the warmer side across both conditions allowed us to separate preference for the warm side from preference for EE. This may also contribute to efforts to identify the proper temperature range in which to present enrichment to maximise its effectiveness. 

Previous experiments on enrichment in snakes have generally used semi-arboreal species such as corn snakes [9] rat snakes [12], or ball pythons (*Python regius*) [13]. No studies that we are aware of have focused on semi-fossorial (burrowing) species, which may require different EE. Young snakes were selected for the current study to represent the numerous hatchlings that are purchased every year, to identify if preference for enrichment is present in the early stages of development. Providing adequate enrichment during development is crucial to the growth of healthy individuals [1].

In addition, the only other study we are aware of to have assessed snakes’ EE preferences [8] used subjects procured from a charity organization, whose individual histories were unknown, and exposed the snakes to EE for less than 24 h to determine preference. Therefore, the aim of the current study was to incorporate a longer exposure to EE (seven days), using subjects of the same age and traceable history to clarify the EE preferences of a popular pet species, the western hognose snake (*Heterodon nasicus*). We hypothesized that snakes would prefer EE over standard areas and prefer warmer over cooler enriched areas. Additionally, we hypothesized that preference for a relatively novel EE may be related to a common measure of boldness—time spent outside of a shelter in a novel environment [28,29]. To test this possibility, we looked for repeatability of boldness across these two contexts. We predicted that if both tests were measuring the expression of a bold-shy personality trait (which has been observed across contexts in, e.g., garter snakes [24]), we would find repeatability between the two tests, driven by high between-individual variability and low within-individual variability.

## 2. Materials and Methods

### 2.1. Subjects and Housing

Subjects consisted of five male and eleven female western hognose snakes, 3–4 months old when acquired from local breeders. The relatedness of the snakes was unknown. In the breeding facility, before arrival to the lab, the snakes were kept in small Tupperware containers with paper towel bedding and a small water dish, meant for easy cleaning. After arrival to the lab the snakes were individually housed in custom-made PVC enclosures (^®^Cornel’s world), that were distributed evenly within the temperature-controlled housing room in columns of 2–3 stacked enclosures. This room was windowless with a single door that was only opened during observational periods by designated researchers. Each enclosure was 46 × 56 × 30 cm. Before the start of the experiment, subjects underwent an acclimation period of one month to minimize effects of stress from travel and introduction to a novel environment. All enclosures had 5 cm deep loose coconut husk substrate (^®^Zoo Med Eco Earth), a single black plastic shelter (14 × 10 × 5 cm; ^®^Cornel’s World) and a water dish (30 × 15 × 5 cm) where fresh water was freely available. These conditions were designed to closely mimic conditions in standard pet snake care [30].

Cages had a strip of LED lighting (2700 Kelvin) along the ceiling on a 12:12 h cycle (lights on at 8:30 a.m.) and belly heat provided by thermostat-controlled heat-tape (THGHeat; ^®^Spyder Electronics HerpStat) under one corner of the cage, which generated a constant heat gradient in the cage (from about 32 °C directly over the tape, to the room temperature of 23 °C). This temperature gradient mimics the gradient found in their natural habitat and used frequently by pet owners and breeders [30,31]. Surface temperatures were monitored daily using a temperature gun (^®^Etekcity, infrared thermometer). In their natural environment, the snakes would have access to both basking heat (from the sun) and belly heat (from sun-warmed rocks or earth), but belly heat provided via heat-tape, is more accurate in maintaining constant temperatures, and mimics standard hognose snake enclosures in the pet trade. Each cage had a 43 cm × 12 cm high sliding glass door in the front. Once a week, snakes were fed in a separate feeding chamber (11 × 15 × 4 cm) placed inside their enclosure. Snakes were fed two pieces of either a rat (bred in house) or defrosted salmon (^®^Great Value) fillet. Both foods were dusted with a reptile calcium supplement (^®^Zilla), and the pieces were slightly larger than the snake’s head. Snakes were housed in the lab under these conditions for five weeks before preference testing began, in the same enclosures.

### 2.2. Preference Testing

For preference testing, each enclosure was divided in half (Appendix A), one side of which was enriched. There was no barrier between the two sides of the cage, and snakes could move freely between the two halves for seven days. Each side included a black plastic shelter. Unenriched areas were barren except for paper towel substrate and a round water dish (11 cm diameter × 4 cm high). Enriched areas were filled with 10 cm deep coconut husk substrate and contained a larger plastic water dish (11 × 11 × 4 cm), one straw ball (5 cm diameter) and a fake plant (6 × 7 × 7 cm).

The location of the snake within the enclosure was observed four times a day, at 10:00, 13:00, 16:00, and 19:00, starting the day after they were placed in the enclosures (6 days of observation). These times were chosen because they reflect when these diurnal snakes are most active. If the snake was visible through the cage door at the time of observation, it was not disturbed. Otherwise, the door was opened and the objects in the enclosure were gently lifted until the snake was located and observed from approximately 30 cm. Locations were recorded as whether the snake was on the enriched or standard side. The order of observations of subjects were conducted in the same manner every day and took five minutes to complete. During this time the snakes were found to be stationary under objects, with minimal movement when found by the observer. Snakes were only handled once during the experiment, on the last day, for feeding. Otherwise, snakes were not moved from their chosen spot in the enclosure.

Since enclosures had belly heat emanating from one corner, we counterbalanced whether the enriched side was the warmer or cooler side, as well as whether it was the left or right side. In all other respects, all enclosures were identical. Four males and four females were placed in enclosures with a warmer enriched side, and seven females and one male were placed in enclosures with a cooler enriched side.

### 2.3. Boldness Assay

Boldness assays closely followed existing methods [23]. Trials were conducted within 1–2 days of feeding, to ensure that hunger did not affect the snakes’ personal boldness. The assay was conducted in a PVC box (43 × 39 × 38 cm). There was one black plastic reptile shelter, identical to the shelter in the home cage, placed against the centre of one long wall. The bottom of the arena was lined with paper towels (Appendix A). Trials were recorded using a webcam (^®^Logitech c920; videos were recorded at 1080p, 30 fps) mounted above the arena. The temperature of the arena floor was checked with a temperature gun prior to each trial and ranged from 25–27 degrees Celsius.

To start the boldness assay, snakes were placed with their heads close to the entrance of the shelter and allowed to slither into the shelter to start the trial. The proportion of the session that the snake spent outside of the shelter was recorded as a measure of boldness [28]. Assays lasted 15 min and each snake was tested once.

### 2.4. Statistical Analysis

We used Bayesian statistics for all analyses. Analyses were run in JASP [32] and R v 4.2.1 [33]. Each snake was given a daily score between 0 and 1 for the proportion of times it was found on the enriched side of the cage. We used a Bayesian repeated-measures ANOVA to test preference for the enriched side, including the effects of day (within-subjects) and temperature (between-subjects; whether the enriched side of the cage was the warmer side, coded as a binary variable). We additionally ran this model with sex as a second between-subject factor. We report Bayes Factors (BF) comparing each model to the null, along with effect size adjectives as suggested by [34]. We also report Bayesian inclusion factors (BFincl), which estimate the main effect of each factor across all models. For the supported main effects (BF > 1), we report the posterior odds of post-hoc tests, corrected for multiple comparisons [35]. To test whether performance on the standard boldness assay (time outside a shelter) and preference for the enriched environment were related to an underlying bold personality type, we tested for behavioural consistency across the two contexts. To do so, we calculated repeatability between the contexts using the *MCMCglmm* package in R. This method was chosen over simply correlating the two test scores as it allows us to report between- and within-individual variability across the contexts. For the analysis, we used weakly informative (ν = 0.002) inverse-Gamma priors for the variances. The model ran 65,000 iterations with a 15000 burn-in and a thin of 50. Along with repeatability values, we report between- and within-individual variability, along with 95% credible intervals (CI; the Bayesian version of a confidence interval). We note that priors for this analysis are bounded at zero. As such, credible intervals near zero should be treated with caution.

## 3. Results

The snakes weighed 12.02 ± 3.38 g at the start of the experiment. We found strong evidence [34] that snakes preferred the enriched side of their enclosures over the standard side (Figure 1; grand mean preference score = 0.68 ± 0.22 SD; one-tailed *t*-test, BF = 18.91). We found no effects of sex on preference (the best model, BF = 6.24, included only effects of day and temperature; main effect of sex, BFincl = 0.58; post-hoc comparison across sexes, posterior odds = 0.45). We therefore exclude Sex from the remaining analyses. We found moderate evidence that preferences were affected by day (BFincl = 3.56) and anecdotal evidence for an effect of Temperature (BFincl = 1.39; full ANOVA results are given in Appendix A). Post-hoc comparisons showed strong evidence for an effect of temperature (odds = 11.72), displaying a preference for the cooler side, and a moderate difference between days 4 and 6 only (odds = 3.16; all other odds < 0.75). However, we also found moderate evidence for a growth in preference across days (Bayesian linear regression: BF = 7.29, β = 0.046, 95% credible interval = [0.00, 0.082]).

Repeatability between enrichment preference and boldness scores was 0.3 (95% CI = [0, 0.64]). Although this value is similar to the average of ectotherm repeatability [19], the wide credibility interval suggests that this value should be treated with caution. Repeatability was a function of both low between-individual variability (σ_b_ = 0.002, 95% CI = [0, 0.06]) and low within-individual variability (σ_w_ = 0.04, 95% CI = [0.02, 0.08]).

## 4. Discussion

Preference tests for enrichment have rarely been conducted in snakes. Our data demonstrate a robust preference for enriched over standard conditions in western hognose snakes. However, contrary to our prediction, preference for EE increased when it was on the cooler side of the enclosure. We also found an increase in preference for EE over days, suggesting that our results are not solely due to neophilia or familiarity with any aspects of either treatment condition but instead reflect stable preferences.

We did not find sufficient evidence to state that preference for EE was driven by an underlying bold-shy personality trait. Although repeatability between our boldness assay and preference for EE was near to the ectotherm average [19], between-individual variability (an important component of personality) was low and the likelihood that the true repeatability was 0 was non-negligible. Boldness is a commonly observed trait in snakes (*Crotalus oreganus* [26]; *Thamnophis sirtalis sirtalis* [25]; *Boa imperator* [22]) and our inability to detect sufficient repeatability may be due to the extended duration of the EE preference assay: over multiple days, boldness-related variability in preference for EE may dissipate, as the risk-free cage environment becomes more familiar.

Though we do not know which specific aspect(s) of the EE we provided drove our snakes’ preferences, we note that one crucial difference between conditions was the availability of deep substrate, which is likely important and enriching for these semi-fossorial snakes. Snakes on the enriched side were regularly found burrowing in the substrate, possibly because this behaviour enhances crypsis, helps to modulate humidity levels, and provides a slight vertical thermal gradient. This level of regulation would not have been achievable on the standard housing side of the cage. In their natural environments, the snakes would be exposed to an abundance of substrate that resembles the substrate used within this study. Western hognose snakes commonly burrow to assist in thermoregulation, to shelter from predators and to hunt [36]. Additionally, they can often be seen ‘rooting’ through loose soil. Although paper towel substrate, as found on the standard side, does provide opportunities for below-substrate sheltering (many snakes were found under the paper towel), paper towel does not provide the same opportunities for naturalistic digging and rooting. As such, preference for the EE may be driven by increased opportunities to engage in more naturalistic stress-reducing behaviours such as digging and rooting.

We also found that snakes preferred enrichment on the cooler side of the enclosure, in contrast with research that suggests heat is rewarding for reptiles [37]. For example, water snakes improved in a maze task when rewarded with a warm shelter [38], and two lizard species displayed decreased latency to goal sites in a maze when rewarded with heated rocks [39]. In these examples, heat was rewarding because subjects were colder than their ideal temperature. Our enclosures mimicked temperature gradients used widely by hognose owners and it has been found that the ideal body temperatures of free-ranging hognose snakes are 27–30.5 degrees Celsius [31]. This indicates that our enclosures were not too hot (Appendix A), especially since we anecdotally observed the snakes frequently resting on the hot side after feeding [40]. This aligns with findings that mention snakes actively resting in hotter temperatures to aid in digestion after feeding [27]. We therefore suggest that hognose snakes, who frequently burrow into cooler layers of their substrate, may not gravitate to heat in the same way as other snake species [41]. Alternatively, snakes may prefer to rest and interact with stimuli in cooler temperatures to reduce their metabolism between feedings [42]. This finding further reinforces the need for EE preference testing with temperature, especially in under-studied species such as snakes, that rely on their environment for thermoregulation.

Preference testing of enrichment prior to implementation is crucial, especially in understudied species, as it maximises the effectiveness of the enrichment, circumventing unnecessary exposure to stimuli that may be aversive or non-beneficial. This is especially true for snakes, given how little we understand of their behaviours and stress responses. Our findings align with previous studies that highlight the need to understand an animal’s species-specific thermal preference, as our hognose snakes preferred EE at cooler temperatures. Understanding the relationship between temperature and other EE is crucial for snakes given their need for a thermal gradient within enclosures. Our results suggest that pairing of heat sources and other EE may encourage or discourage the use of said EE depending on the appropriateness of the pairing. Future research should test hognose snakes with overhead heating rather than below substrate heating which may interfere with their ability to borrow for cooling down. Our results also highlight the importance of preference testing as a stress-free observational tool that provides subjects with the freedom to choose, and thus should be used more often in studies of reptilian enrichment.

We note that along with any aforementioned limitations, we were unable to acquire an equal number of males and females and thus have a sex bias towards females. Additionally, any future preference testing should pseudo-randomize the order of observation to minimize any anticipatory behaviours.

## 5. Conclusions

In conclusion, western hognose snakes prefer environmental enrichment, and our findings highlight the effectiveness of using preference testing with cryptic species such as snakes. With their complete behavioural repertoires still unknown, it is difficult to assess which resources they benefit from. However, by providing the snakes with a choice between different resources, and observing which ones they allocate the most amount of time to, we can be confident about which resources to implement in the long term. This method minimizes stress on subjects and allows us to include resources that are sought after, as well as increasing available space by removing unnecessary stimuli. We also found that burrowing snakes tend to prefer enrichment when provided at lower temperatures within their acceptable range. This further encourages the need to preference test resources when observing ectothermic animals to identify the optimal manner in which to present enrichment.

## Figures and Tables

**Figure 1 animals-12-03347-f001:**
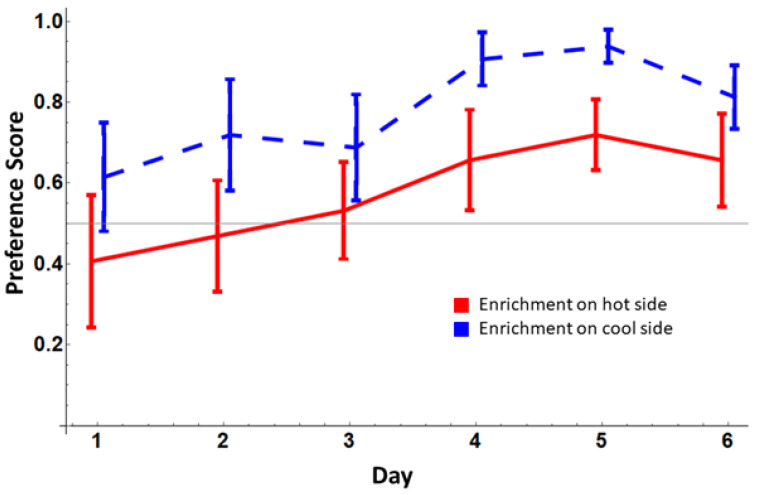
Preference for the enriched side of the enclosure as a function of day (x-axis) and whether the enriched side of the cage was warmer (solid red line) or cooler (dashed blue line). Error bars show ± SEM.

## Data Availability

All the data analyzed in this paper are archived at https://osf.io/cpf97/ (accessed on 29 September 2022).

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
