# Peer review of "Western Hognose Snakes (Heterodon nasicus) Prefer Environmental Enrichment"

_animals, 2022, doi:10.3390/ani12233347_

Round 1

Reviewer 1 Report

The manuscript explores the preference for environmental enrichment in hognose snakes. For a person who takes care of snakes, it sounds very interesting. The sadder was the discovery that the authors called an enriched environment one, which was standard for me. Unfortunately, it is common in the pet trade to keep snakes only on paper towels. From this point of view, this manuscript is very important.

The manuscript is quite clearly written. I missed several references from the text in the final list (e.g., Bechard et al. 2011; Hollandt et al. 2021; Almli and Burghardt 2006; Manrod et al. 2008; Jolles et al. 2016; R Core team 2022).

I have a problem with the design of the study. I do not understand why the authors complicated the study with the temperature. At first, they did not mention the species' preferred temperature, which is crucial. Second, they did not show the temperatures in the cage properly. It is necessary to show the data about the gradient in the enclosure at least in the supplementary material. Nevertheless, I suggest: do not complicate your study with temperature. It is a different issue.

The second problem is with personality. It can be interesting if it will be measured properly. Nevertheless, it was not the case. The behaviour of snakes cannot be repeatable when they pass the test only once. Moreover, you need to measure it in various contexts to call it personality.

Row 192: I suggest calling the factor “Temperature” rather than “Hot”  

Row 310: it is not necessary to have b in reference Skinner et al. 2022.

Reviewer 2 Report

Review report

Western Hognose Snakes (Heterodon nasicus) Prefer Environmental Enrichment

Summary

This paper presents the results of a preference test in Western Hognose Snakes. Individuals were divided into 2 treatment groups (enriched-hot, barren cold vs. enriched-cold, barren-hot), and their preference was assessed by measuring the time spent in each side of the enclosure, estimated by a scan sampling method conducted 4 times a day, over 7 days. The findings of the study indicate that these snakes prefer enriched sides of the cage over barren sides, and that this preference was more evident when the enriched side was cooler. The analysis is complemented by measures of intra- and inter-individual variability and repeatability to infer whether personality could influence the results.

I would like to congratulate the authors for this research. I sincerely hope my comments can help improve the communication of their work.

General concept comments

Is the manuscript clear, relevant for the field and presented in a well-structured manner?

In my opinion, the manuscript is extremely relevant for the field. Millions of reptiles kept in captivity suffer significant welfare impacts throughout their lives, partly due to knowledge gaps related with their environmental requirements. Preference tests are an excellent way to gather knowledge on reptiles’ environmental preferences, and should be more routinely reported. Their major strength lies in the fact that (irrespective of whether the animal benefits from its decision in terms of health or fitness) the preference demonstrated can be expected to be associated with subjective experiences (seeking choice that makes it “feel good” or avoiding the choice that makes it “feel bad”), and is therefore tightly linked to the animal’s welfare. This research provides a simple and elegant example, with rather surprising results. The article is well structured, clear, and objective in the assessment of the core hypotheses. In terms of overall flow, the title, abstract results, discussion and conclusion are well aligned.

Are the cited references mostly recent publications (within the last 5 years) and relevant? Does it include an excessive number of self-citations?

The article includes a comprehensive set of the most relevant and recent references for the topic, with no unnecessary self-citations.

Is the manuscript scientifically sound and is the experimental design appropriate to test the hypothesis?

The experimental design of the manuscript is generally sound. However, some aspects could benefit from clarification: the conditions of the pre-study period are quite different from the barren side, but quite similar to the enriched side, especially in terms of substrate. This should merit consideration from the authors. Specifically, could familiarity make the animals choose the enriched side more frequently, or avoid the barren side? Additionally, no information is given about the order by which observations were made in each round, and at what distance the observer was. In case the observations were not randomized, an influence from observer presence is quite possible, and perhaps even expected in young animals housed in a lab and fed once a week for five weeks. Ideally, both the terrarium locations within the room and the observation order should be randomized to avoid bias (See ARRIVE guidelines). Additionally, there is a sex bias in one of the groups. The possibility of it influencing the results should be mentioned/explored.

Are the manuscript’s results reproducible based on the details given in the methods section?

The manuscript would benefit from some additional methodological details regarding the animals (are they all siblings or genetically related?), the enclosures (are enclosures randomly allocated to rack or room position? Light, distance to people, windows and doors can all influence animals’ behaviour. Is feeding or maintenance in enclosures performed in the same or random order?), observation (what was the order of observation among enclosures? this can influence the results by the stimulation of anticipatory behaviours).

Are the figures/tables/images/schemes appropriate? Do they properly show the data? Are they easy to interpret and understand? Is the data interpreted appropriately and consistently throughout the manuscript?

The single figure is appropriate, clear and easy to interpret.

Are the conclusions consistent with the evidence and arguments presented?

The conclusions are quite factual and objective, and adequately contextualized with the literature. Some more integration with the species natural biology could be provided (for example, regarding the temperature gradient, what is known of the temperatures in the species' range? is fossorial behaviour used for thermoregulation in this area? to avoid heat? cold? both?). Drawing links to each species' natural history when designing captive environments should be encouraged. Although it is implicit in the rationale of the study, explicitly including considerations of the species’ natural history in the discussion would enrich the manuscript.

Potential limitations are also insufficiently discussed. Limitations that should be discussed are: sex bias, observation order and terrarium location within room, and the similarity of pre-study conditions to the enriched side.

Please evaluate the ethics statements and data availability statements to ensure they are adequate.

I couldn’t find an ethics statement. This manuscript used live animals under captive (inevitably sub-optimal) conditions, and therefore requires an ethical review that needs to be mentioned. Further, the manuscript could be improved by adding a short paragraph containing a) a short explanation of the choice of the animals (e.g. young, vs. adults); b) mention of how the potential benefits of this research outweigh the welfare costs to the animals used, thus making animal use acceptable; c) mention of the determination of sample size and statistical power to enable meaningful results with the least number of animals; d) a brief description of the refinements that were implemented where possible to improve the animals welfare during the study; e) the destination of the animals after study conclusion.

Specific comments (by line number)

Line 28: saying that unfit housing stems from the lack of environmental enrichment does not seem correct here, especially because you go on to provide two examples where basic needs are not met by the characteristics of the environment. The question then arises whether providing basic environmental needs constitutes enrichment? Enrichment is a difficult concept to manage in these discussions, as its definition is not straight-forward. Indeed, it can be seen as Burghardt (2013) argues, as a way to mitigate environmental deprivation.

Line 40: again, what needs researching, environmental enrichment needs or environmental and behavioural needs of reptiles? Ideally, if they have adequate environments, they won’t need EE. I would advise caution in the use of the expression “EE needs”. Animals don’t specifically need EE, they need adequate environments, and when not possible we can use EE to mitigate the impacts of the environmental deficits.

Lines 115 and 139: the conditions in the pre-study period are remarkably similar to those of the enriched side of the enclosure regarding substrate and water dish. Could this cause a familiarity bias in the results? Are they being faced with a choice between an enriched and barren environment or between a familiar and unfamiliar substrate/setting? There could be an issue here between a preference test and a novelty test.

Line 142: In which order were the observations made? There could be an influence of observer presence if the order was not randomized. For example, if observations were always made in the same order, perhaps the later snakes would be less likely to be found burrowing, and could approach the glass or even change sides. This is very common with reptiles that are only disturbed to be fed by humans. Please provide clarification, and even if there was no randomization, provide some clarification on what order was used to collect the observations, and whether movement of the animals was observed in response to human presence.

Line 153 and 231: males and females were not distributed evenly (approx.) between treatment groups. The cold group is almost exclusively female. Is there a chance that the preference in the cold group is due to a sexual difference in preferred temperatures?

Reviewer 3 Report

Dear editor,

This study is a short observation of the use of enriched shelters in captive snakes. The results graphically showed  the interaction among several already demonstrated facts in the literature. My only two concerns are:

1)“shelter type” (sand vs paper) seems confounded with “enrichment”. The authors need to explain why this is not the case or discuss the implications of this problem for the conclusions taken from their experiment.

2) the authors put care into separating what are new conclusions from this observation and what is supporting already demonstrated facts from previous literature. I not only mean the ones cited by me along my comments but after a thorough literature review.

3) the raw data and analysis scripts should always be provided to ensure repeatability and evaluation by the readers.

Below, I provide more specific comments to help make the note’s arguments stronger and more in line with what is already known about thermal preferences in snakes and other animals.

Abstract

Ectothermic animals do not always prefer hotter temperatures, but temperatures in certain ranges that enclosures need to ensure, see on ways to identify these conditions for reptiles. For example:

Camacho, A., & Rusch, T. W. (2017). Methods and pitfalls of measuring thermal preference and tolerance in lizards. Journal of Thermal Biology68, 63-72.

Explain which variables represented “enrichment” here. Because it seems that enrichment is confounded with the type of shelter in this experiment, and it should not be.

90-97. Please rephrase to make the paragraph’s style “introductory” and not “methodological”, as it stands now.

90-91. Heat is a physical variable, avoid confusion among the different potential dimensions of EE.

103-104. why knowing is that important? That should be explained if it is or the argument removed if it really doesn’t matter.

130: what is “Great Value”? Please avoid localisms in an international journal. If it is a bradn add the ® symbol.

134.

156. These assays must have been done in the exact same temperature, as temperature affect the flight or fight behavior of reptiles. Can that be granted?

Keogh, J. S., & DeSerto, F. P. (1994). Temperature dependent defensive behavior in three species of North American colubrid snakes. Journal of Herpetology28(2), 258-261.

158. this should say “affect snakes’ personal boldness ”, but then that is hard to assess , right?

239. It has been actually observed that snakes look for hot environments after feeding which helps for a fast digestion.

Sievert, L. M., & Andreadis, P. (1999). Specific dynamic action and postprandial thermophily in juvenile northern water snakes, Nerodia sipedon. Journal of Thermal Biology24(1), 51-55.

243. This has been already suggested before and thus that should be acknowledged.

250. No, actually the dozens of studies on thermal preference of snakes, reptiles and so many other animal groups have long demonstrated that animals have species’ specific thermal preferences.
